# Shaping the Physicochemical, Functional, Microbiological and Sensory Properties of Yoghurts Using Plant Additives

**DOI:** 10.3390/foods12061275

**Published:** 2023-03-17

**Authors:** Joanna Wajs, Aneta Brodziak, Jolanta Król

**Affiliations:** Department of Quality Assessment and Processing of Animal Products, Faculty of Animal Sciences and Bioeconomy, University of Life Sciences in Lublin, Akademicka 13, 20-950 Lublin, Poland

**Keywords:** yoghurt, plant additive, nutritional value, bioactive compounds, probiotic strains, pro-healthy properties, innovations

## Abstract

Nowadays, consumers pay particular attention to the quality of the products they buy. They also expect a high level of innovation. Hence, the offer from the dairy sector is increasingly focusing on the use of various additives with proven health benefits. Many scientific teams from various regions of the world are engaged in research, and their aim is to identify plant additives that have beneficial effects on the human body. The aim of this article was to summarize the latest literature pertaining to the effects of plant additives used in the production of yoghurts on their physicochemical, functional, microbiological and sensory properties. It was found that a wide range of additives in a variety of forms are used in the production of yoghurts. The most common include fruits, vegetables, cereals, nuts, seeds, oils, plant or herbal extracts, fruit or vegetable fibre, and waste from fruit processing. The additives very often significantly affected the physicochemical and microbiological characteristics as well as the texture and sensory properties of yoghurt. As follows from the analysed reports, yoghurts enriched with additives are more valuable, especially in terms of the content of health-promoting compounds, including fibre, phenolic compounds, vitamins, fatty acids and minerals. A properly selected, high quality plant supplement can contribute to the improvement in the generally health-promoting as well as antioxidant properties of the product. For sensory reasons, however, a new product may not always be tolerated, and its acceptance depends mainly on the amount of the additive used. In conclusion, “superfood” yoghurt is one of the products increasingly recommended both preventively and as a way of reducing existing dysfunctions caused by civilization diseases, i.e., diabetes, cancer and neurodegenerative diseases. The studies conducted in recent years have not shown any negative impact of fortified yoghurts on the human body.

## 1. Introduction

For thousands of years, milk has been a permanent staple in the human diet. The first known attempts to obtain milk from cattle date back to around 15,000 BC, as evidenced by wall paintings in caves. The accidental discovery of the lactic fermentation process in ancient times only refined this unique raw material, giving rise to new dairy products such as yoghurt, kefir and buttermilk. These days, yoghurt has become a globally recognised commercial product with more and more consumers buying it every year [1,2]. Table 1 shows the scale of yoghurt production globally, in Europe and in Poland [3].

Yoghurt is a dairy product characterized by a very simple yet interesting composition, containing many compounds with health-promoting properties. It is an oil-in-water (o/w) food emulsion; the oil phase consists of fatty droplets, and the water phase is a solution of proteins, sugars and mineral salts. What distinguishes yoghurt from other dairy products is the presence of characteristic cultures of lactic acid bacteria, typically *Streptococcus thermophilus* and *Lactobacillus bulgaricus*, producing lactic acid, which, thanks to it lowering the pH of the food matrix, hinders the transmission of pathogens in food [7].

Due to the scale of milk production, yoghurt is most often made from cow milk, but can also be made from buffalo, sheep, goat or camel milk [8,9,10]. The species from which the milk is obtained affects the nutritional value and physical characteristics of natural yoghurt (Table 2). Yoghurts made on the basis of sheep and buffalo milk are characterized by the highest content of total protein (6.2 and 4.7%, respectively), and in sheep, also of fat (7.0%). Importantly, goat milk products are the poorest source of lactose (3.3%).

Yoghurts contains some of the most nutritious ingredients with considerable health benefits. During the fermentation process, bioactive peptides and bacteriocins that reduce the content of lactose in products are also released, which is extremely important from the point of view of people with lactose intolerance [22].

Natural yoghurts are characterised by unique sensory properties. A slightly sour taste and a pleasant, gentle smell provide an excellent base for creating new dairy products and improving the existing ones. A wide range of fermented milk products can be found on store shelves. In recent years, health awareness has increased the consumer demand for natural, fermented milk products, as well as products fortified with plant additives, which are a valuable source of health-promoting ingredients. Manufacturers are constantly expanding their commercial offer by enriching yoghurts with various additives, ranging from fruit and vegetable bits to nuts and dried fruits, herbs, cereals and flowers [23,24,25,26]. Due to their nutritional value and health benefits (thanks to the content of calcium, phosphorus, B vitamins and other compounds depending on the additive used), as well as the presence of specific microflora, the popularity of fermented milk products is constantly growing [4,27,28].

The aim of this study was to present the impact of plant additives on the physicochemical, functional, microbiological and sensory properties of yoghurts. To this end, a review of the available literature was performed, focusing on reports from the last 5 years.

## 2. Materials and Methods

A comprehensive search for scientific articles on the reviewed topic, i.e., pertaining to the physicochemical, functional, microbiological and sensory properties of yoghurts containing plant additives, was carried out using databases such as Scopus, Science Direct, Springer, PubMed, Medline and Web of Science. When searching for information, the following keywords were used: “yoghurt”, “plant additive”, “properties”, “nutritional value”, “texture”, “acidity”, “sensory properties”, “syneresis”, “bioactive compounds”, “innovation”, “fruit”, “vegetable”, “nut”, “seed”, “herb”, “tea”, “flower”, “probiotic strains” and “health”. The search was mainly limited to reports in the English language. Based on the criteria and keywords listed, 199 reports (including 181 research studies) were selected for further analysis. The review included works published in the years 2000–2023, of which 70% were from the last five years, i.e., 2018–2022. Additionally, the Appendix A is provided, i.e., Appendix A showing PRISMA 2020 flow diagram and Appendix A—PRISMA 2020 Checklist.

## 3. Physicochemical Changes in Yoghurts

Yoghurt is a fermented milk product in which the presence of fruit, vegetables or other plant components can significantly affect the measurable values of chemical parameters (fat, protein, ash, dietary fibre, minerals and vitamins) and physical parameters (moisture, acidity or syneresis). This is indicated by research conducted over at least the last 10 years. Saeed [29] showed that the addition of 2% *Moringa oleifera* leaf powder to yoghurt increased the content of protein (from 4.11 to 4.68 g/100 g), lactic acid (from 0.871 to 1.137%) and ash (from 1.21 to 4.21 g/100 g). On the other hand, when comparing the active acidity based on the pH value, or moisture, significant differences were not observed. Moreover, the addition of dried leaf powder was characterized by a high content of crude fibre (19.2 g/100 g), calcium (2003 mg/100 g), phosphorus (204 mg/100 g) and potassium (1324 mg/100 g), which resulted in the value of yoghurt.

The fortification of yoghurt with plant additives determines the content of their characteristic ingredients in the product [30]. In turn, Hussein et al. [31] assessed the effect of adding chickpea flour to bio-yoghurt. They showed that a 3% addition reduced the protein content by more than 8%, i.e., from 4.35% in the control yoghurt to 4.0% in the product containing 3% of the additive. There was also an increase in the amount of fat from 3.14% in the control yoghurt to 3.3% in the fortified yoghurt, and the lactic acid content from 0.74% to 1.21%, respectively. The authors included the effect of the storage period. The storage period additionally reduced the pH value from 4.63 on the first day to 4.22 on the 21st day of yoghurt storage and, respectively, from 4.41 to 3.91 in the yoghurt with a 3% additive. On the other hand, Bulut et al. [32] showed that added plant protein increased titratable acidity values during storage. Abdel-Hamid et al. [33] assessed the variability in the physicochemical parameters of yoghurts with the addition of *Rubus suavissimus* S. Lee Chinese sweet tea extract. They showed that a 1% addition of the extract increased the protein level compared to the control yoghurt from 5.1% to 5.18%. Similar dependencies were observed by Van Nieuwenhove et al. [34] who assessed yoghurt fortification with jacaranda seeds (*Jacaranda mimosifolia* D. Don). They showed that a 0.5% supplement increased the amount of protein from 3.46% in the control yoghurt to 3.53% in the supplemented yoghurt. However, the addition did not affect the fat content, which in both cases was 1.43 g/100 g but was, in turn, influenced by the storage time. On the 28th day of storage, the fat level in the control yoghurt decreased by 5%, amounting to 1.48 g/100 g, and in the case of yoghurt with the addition, it more than doubled—to 3.55 g/100 g. On the other hand, the pH of the yoghurt fortified with jacaranda seeds decreased (from 4.51 to 4.41). In the control yoghurt, the pH value decreased comparably, i.e., from 4.51 to 4.42.

Yildiz and Ozcan [35] assessed the effect of adding various vegetable pulps to yoghurt obtained from fresh cow milk. The addition of individual vegetables (pumpkin, carrots, green peas and zucchini) in the form of puree increased the amount of dietary fibre (amounting to, respectively: 1.05%, 0.55%, 1.12%, 0.67%). The addition of pumpkin fibre (in the amounts of 0.5, 1.0 and 1.5%) was also used by Bakirci et al. [36]. In each of the yoghurts, the amount of fat increased, respectively, by 0.02, 0.03 and 0.04%. These changes may be due to the fat content of the plant raw material itself and reduced syneresis [37]. Al-Shawi et al. [38] analysed yoghurts with the addition of an aqueous and alcoholic thyme extract over a 28-day storage period and observed an increase in the amount of fat from 0.184% to 0.360% in yoghurt containing the alcoholic extract, and from 0.177% to 0.326% in yoghurt with the aqueous extract, compared to an increase from 0.178% to 0.32% in the control. A similar relationship was demonstrated by Felfoul et al. [39] who observed that with the increasing addition of ginger (*Zingiber officinalis*), introduced at 0.5, 1.0, 1.5 and 2.5%, the fat content also increased (respectively, to: 5.38, 6.0, 8.17 and 9.91%) compared to the control yoghurt with the fat level of 3.82%. Hasneen et al. [40] showed that the addition of 10% water extracts of turmeric (*Curcuma longa*), sage (*Salvia officinalis*) and marjoram (*Origanum majorana*) increased the pH value (respectively: 4.66, 4.99 and 4.82) and the content of dietary fibre (respectively: 0.23, 0.43 and 0.33%) relative to the control yoghurt, where the pH value was 4.71 and the fibre was 0%. There was a decrease in the content of protein (3.85, 3.96 and 3.89%), fat (1.1, 1.2 and 1.1%), dry matter (12.84, 12.75 and 12.61%) and lactic acid (1.17, 0.85 and 0.91%) as compared to the control, where protein was 4%, fat—1.20%, dry weight—12.92% and lactic acid—1.12%.

Acidity is an important parameter determining the resumption of the product. In yoghurts containing plant additives, i.e., zucchini, carrot, pumpkin, inulin or other source of soluble dietary fibre, an increase in active acidity (expressed as a decrease in the pH value) can be observed, also during storage [41]. Szołtysiak et al. [42] assessed the variability in the acidity in yoghurts fortified with honeysuckle berries (*Lonicera caerulea* L. var. Kamtschatica), the effect on singlet oxygen quenching, as well as the possibility of chelating metal ions. They demonstrated that the pH of yoghurt with the addition of fruit was 4.55 and the lactic acid content was 1.09%. In the control yoghurt, the values were under 4.51 and 0.98%, respectively. In turn, Dabija et al. [43] showed that enrichment of yoghurt with wild herbs (0.25%), i.e., thistle (*Carduus* L.), hawthorn (*Crataegus* L.), sage (*Salvia officinalis* L.) and marjoram (*Origanum majorana* L.) increased the level of acidity. They also considered the effects of 28-day storage. The initial acidity (T) for the respective additives was thistle—101, hawthorn—100, sage—110 and marjoram—107. After the analysis on day 28, the acidity level increased for thistle by 15% and hawthorn—5%, whereas it decreased for sage by 4% and remained unchanged for marjoram. Yekta and Ansari [44] studied the effect of jujube mucus at various concentrations (0.1, 0.15 and 0.2%) on physicochemical changes resulting from the amount of plant additive used. They demonstrated that the addition of fruit mucilage reduced the pH from 4.07 to 3.9, thus increasing the lactic acid content from 0.96% to 1.06%. When analysing the acidity of yoghurts with the addition of a corn starch, Zarroug et al. [45] observed that the higher the additive content, the lower the acidity (expressed in °D and as pH value) compared to the control yoghurt. This may be due to a decrease in the amount of water in the yoghurt, which consequently leads to a reduced lactose fermentation by probiotic bacteria, ultimately resulting in inhibition of lactic acid production.

The addition of acidic plant products has a significant effect on the occurrence of syneresis. Fruit or vegetables added to yoghurt increase its dry matter and the amount of pectin, which increases water absorption and thus reduces syneresis. There is also a noticeable decrease in viscosity, mainly due to the acidity of fruit additives. However, Sengul et al. [46] observed an inverse relationship in their research. The addition of cherries to yoghurt caused an increase in viscosity during storage, which may have been due to the high acidity of the fruit itself. In addition, the weakening of the curd due to casein aggregation and the increase in acidity caused by lactic fermentation led to the emergence of visible whey on the surface of yoghurt [47,48]. Yekta and Ansari [44] showed that the enrichment of yoghurt with jujube mucus in various concentrations (0.1, 0.15 and 0.2%) reduced the degree of whey separation while increasing the water retention capacity. Similar changes for whey syneresis were obtained by Hassan et al. [49] after adding cress mucilage to yoghurt, and by Basiri et al. [50] after enriching yoghurt with linseed mucilage.

Selected studies on the yoghurts with the plant-based additives are presented in Table 3.

However, it should be noted that research studies on yoghurt-like plant products (vegan yoghurts) are beginning to appear in the literature, covering the physicochemical, microbiological and sensory properties [72].

## 4. Microbiological Changes in Yoghurts

Yoghurt fermentation is due to the use of lactic acid bacteria (LAB), specifically *Streptococcus thermophilus* and *Lactobacillus delbrueckii* subsp. *bulgaricus*, in their production. What matters is that the bacteria remain alive, in large numbers (at least 10^6^ cfu/g) until the last day of use [73,74]. The formation of organic acids, including lactic acid, increases the safety of the product through the acidification of the raw material. The formation of other metabolites, e.g., ethanol, enzymes, aromatic compounds or bacteriocins and exopolysaccharides, is responsible for the increased stability, texture and nutritional properties of natural yoghurt. However, the plant additives used should be of appropriate, high microbiological quality so as not to disturb the balance in the starting yoghurt and not to constitute a source of microbiological hazards in the form of bacteria, mould or fungi [75,76].

In yoghurts containing various types of additives, such as fruits, vegetables, legume seeds, omega-3 acids, spices, herbs, grains, mucilage, extracts, nuts and oils, the amount of lactic acid bacteria ultimately changes. Unfortunately, in most cases, these changes are unfavourable, leading to a reduction in the number of living bacteria below the required standard. Such changes are controversial, which is why extensive research is carried out on microbiological changes in yoghurts as a novel food [77,78,79,80,81]. The literature describes cases where the presence of certain plant additives such as persimmon, mango or guava increased the amount of LAB in yoghurts on the 7th day of storage [82].

Fermented milk products are foods that are particularly sensitive to time and conditions of storage. In order to maintain a high-quality product without physicochemical, microbiological, sensory or organoleptic changes, yoghurts should be properly stored, because the variability in LAB bacteria is extremely important in the context of the product’s functional characteristics [83]. In order to determine the microbial variability in yoghurts with respect to storage time and temperature, Zhi et al. [84] conducted studies in which they showed that the longer the storage time and the higher the storage temperature, the greater the degree of changes observed in yoghurts. An initial increase in the number of probiotic bacteria was observed, followed by a decrease in the number of LABs over time.

The addition of natural sweeteners in the form of various types of honey, e.g., pine honey, has a beneficial effect on the microbiological profile of yoghurts [85,86,87,88,89]. Mohan et al. [90] showed that in addition to the satisfactory taste, the presence of honey, in particular manuka AMF15, had antibacterial properties and promoted the growth of fermentation metabolites, especially lactic and propionic acid. In addition, this honey was characterized by the highest content of probiotic bacteria after 3 weeks of refrigerated storage compared to yoghurts with the addition of other honeys (Manuka Blend and UMF18), it also improved the apparent viscosity of the yoghurt.

It has also been reported that the addition of fruit (bananas, strawberries, apricots, peaches) and vegetables (pumpkin, peppers, fennel, tomatoes) increases the content of probiotic and prebiotic bacteria and facilitates other health benefits [37,91,92,93]. Senadeer et al. [94] investigated the effects of various types of pulps made from the *Annona* family fruit, commonly known as graviola. They showed that adding fruit pulp to yoghurt enhanced the growth and viability of *Bifidobacterium animalis* subsp. *lactis* BB–12 during storage, which did not affect the base yoghurt microflora.

Microbiological changes in fermented milk products may occur due to the presence of plant-derived fibre. The fortification of yoghurts with fruit or vegetable fibres, which are phytochemical antioxidants, increases the content of alpha-linolenic acid and is an appropriate matrix for the growth and function of the intestinal microflora, i.e., it improves proteolytic activity leading to the formation of amino acids [95]. Karaca et al. [96] enriched fat-free probiotic yoghurt with apricot fibre, in various concentrations (1 and 2%). The addition of fruit fibre increased the number of *Lactobacillus delbrueckii* subsp. *bulgaricus*, *Lactobacillus acidophilus* LA5 and *Bifidobacterium* BB-12, and the most favourable configuration was found to be *Lactobacillus acidophilus* LA5 as a probiotic culture with the addition of 1% (*w*/*v*) apricot fibre.

Functional additives that affect the microbiological quality of yoghurts can also include herbs (green tea), spices (cinnamon, rosemary, vanilla) and flowers (hibiscus, cornflower, rose) or algae (spirulina) [97,98,99]. Their addition significantly increases the content of added probiotic bacteria cultures in yoghurts. Their antioxidant profile is also growing, which adds to the protection of the human body against civilization diseases [100].

Furthermore, the addition of inulin to natural yoghurts promotes the growth of the desired microflora, thus creating a symbiotic yoghurt with appropriate texture and sensory properties [41]. Moreover, researchers have assessed the effects of less popular additives, e.g., almond milk [101], pinecones [102], olive leaf hot water extract [103] or pullulan [104], on shaping the microbiological quality of the yoghurts produced.

The general direction of changes in the microbiological quality of yoghurts—expressed as the total count of lactic acid bacteria—depending on the plant additive used and storage time, is shown in Figure 1. It illustrates the microbiological changes that occur in the yoghurts during storage. Each additive is marked with a different colour. The y-axis shows the count of bacteria, and the x-axis shows the days on which the analyses were performed.

## 5. Sensory Changes in Yoghurt

Significant differences can also be noticed in terms of the sensory characteristics (taste, smell and appearance) of yoghurts fortified with plant additives. The final evaluation of the finished product depends on the external appearance of the additive itself and the quantity in which it was used. Typically, the sensory evaluation of the product tends to deteriorate with the increased presence of additives. It is therefore important to conduct test production to optimize the amount of the additive when designing a new product. Available reports indicate that enriching yoghurts with herbs or green parts of plants significantly deteriorates their taste. Reduced palatability, consistency and appearance negatively impact the overall sensory evaluation [105]. Külcü et al. [106] determined sensory changes in yoghurt with the addition of *Mentha pulegium* L. (pennyroyal) powder at concentrations of 0, 0.05, 0.10, 0.15 and 0.20%. A 5-point hedonic scale was used to evaluate the yoghurt (1: least acceptable, 5: very acceptable). The control yoghurt received 5 points/5 points max, while the yoghurt with 0.2% addition only 2 points/5 points max, and the difference was statistically confirmed. The smell of yoghurt containing the powder was also less acceptable for the evaluators (5 points for yoghurt with no addition and 4 points for yoghurt with 0.2% addition). The colour of the yoghurts was assessed equally high (5 points) for each sample. The mouthfeel was the best for natural yoghurt (5 points), while yoghurt with 0.15 and 0.20% pennyroyal was evaluated as the worst. When assessing sweetness, the majority of panellists rated the control yoghurts at 3 points, and the yoghurt with the highest amount of mint powder was the sweetest. Keshavarzi et al. [107] analysed the effects of adding ethanol extract (0.2 and 0.4%) and essential oil (0.01 and 0.03%) of *Ferulago angulata* on the sensory properties of yoghurts. Yoghurts containing 0.40% extract and 0.03% oil had a better mouthfeel, comparable to the control yoghurt, than the other yoghurts. In terms of overall acceptance, the yoghurt containing 0.03% of *Ferulago angulata* essential oil was rated the highest.

In turn, Świąder et al. [108] assessed the effects of four different types of teas as functional yoghurt additives. The study used 1, 2, 4, 6 and 8% extracts of green tea, black tea, oolong tea and lemon balm. It was shown that the most acceptable products were plain yoghurt and yoghurt containing 2% oolong tea. Each evaluated parameter, i.e., smell, taste, general appearance and general acceptability of the produced yoghurt, differed in their statistical significance (*p* ≤ 0.05). The intensity of each attribute was measured using a linear unstructured 10-point scale (cu—arbitrary units), from “0”—the worst to “10”—the best perception. Considering the smell, its acceptability decreased as follows: yoghurt with oolong tea (5.9 cu), control yoghurt (5.7 cu), lemon balm (3.8 cu), black tea (3.7 cu) and green tea (3.0 cu). Yoghurt with the addition of oolong tea (5.6 cu) and the control yoghurt (5.3 cu) proved the most acceptable, and therefore the most commercially viable, while significantly lower willingness to buy was indicated for yoghurts containing lemon balm (3.3 cu), black tea (3.2 cu) and green tea (2.3 cu).

When analysing the sensory properties of yoghurts with the addition of fruit and vegetables, it should be emphasized that the overall assessment is very diverse. Öztürk et al. [109] evaluated the addition of husked olive flour (*Elaeagnus angustifolia* L.) and unshelled olive flour at the concentrations of 1% and 2%. The results showed that increasing the content of olives worsened the overall rating of both yoghurts from 4.67 points/5 points max for the control yoghurt to 3.0 points for yoghurt with 1% husk flour and 4.33 points for yoghurt with 1% addition of flour from unshelled olives. The addition of sea buckthorn was analysed by Terpou et al. [26] and turned out to be quite acceptable for the panellists. The overall rating of yoghurt containing sea buckthorn was much higher than that of the control yoghurt. At the same time, the results reported by Brodziak et al. [110] for yoghurts with the addition of sea buckthorn were contrary to those presented by Terpou et al. [26]. The statistically significant (*p* ≤ 0.05) highest scores were given to yoghurt without the addition of sea buckthorn, i.e., 5 points/5 points max for colour, consistency, flavour and aroma. These results translated into a general acceptance note, as yoghurts without the additive scored at 4.81 points/5 points max, and with mousse—4.54 points. Nonetheless, the authors emphasized that the results obtained for the yoghurts containing sea buckthorn mousse were relatively high—above 4 points/5 points max for each characteristic. This indicated high acceptance of the fortified products.

The colour of yoghurt is an extremely important quality that is directly affected by vegetable additives. The respective changes depend on the colour of individual additives and the ability to oxidize during storage. Barakat and Hassan [91] demonstrated that the addition of pumpkin pulp decreased the attractiveness of yoghurt colour by 15%, while the addition of chokeberry juice (*Aronia melanocarpa*) had the opposite effect and increased colour acceptability [111].

The literature also includes studies discussing the sensory properties of yoghurts fortified with seeds or nuts. Ardabilchi Marand et al. [112] assessed the variability in the sensory parameters in yoghurts with the addition of linseed powder in concentrations of 1, 3 and 5%. They found that the colour became more brownish yellow as the amount of linseed increased. The yoghurt with the most attractive sensory features, i.e., taste, aroma, texture and overall appearance, with an overall rating of 6.55, turned out to be the product without the addition of linseed. Yoghurts containing 1 and 3% of the additive took second place. The lowest scores were given to yoghurt with a 5% addition of linseed. The study found that the addition of linseed alone lowered the overall acceptance (*p* < 0.001) due to increased graininess, the presence of a linseed and oily flavour, and a less attractive colour. Similar conclusions were drawn by Bialasova et al. [113] who demonstrated that the addition of linseed negatively affected the taste and smell of yoghurt. In turn, Vanegas-Azuero and Gutiérrez [114] assessed the effect of enriching yoghurts with 4% ground sacha inchi seeds (*Plukenetia volubilis* L.) and 0.5, 1.0 and 1.5% of β-glucans from *Ganoderma lucidum*. It was shown that all the yoghurts were approved by at least 70% of the evaluators. The yoghurt with no additives (93.3%) received the best scores, followed by the yoghurt with a 4% addition of seeds and 1% beta-glucans (88.7%), the yoghurt with 4% added seeds and 1.5% beta-glucans (84.9%), and the yoghurt with 4% addition of seeds and 0.5% beta-glucans (83.0%). The lowest score was recorded for the yoghurt containing 4% of seeds but without the addition of beta-glucans (71.7%), which suggests that their presence may weaken the sensory characteristics of yoghurts.

Figure 2 shows an exemplary effect of plant additives used on the overall positive acceptability of yoghurts. It is presented in the form of a timeline. The axis shows the maximum amount of the specific additive that was accepted in yoghurt by the testers. An acceptable addition did not exceed 5%. Higher amounts were rejected due to the negative effect on the yoghurt characteristics, for example, unsuitable colour or consistency, unattractive smell or high syneresis.

Numerous studies have been undertaken in this area; however, similar trends have be observed. The summary of the effect of plant additives on the physicochemical, microbiological and sensory characteristics of yoghurts is presented in Table 4.

Additives change the properties of yoghurts. Compared to the plain yoghurts (Table 5), yoghurts containing additives undergo more changes during storage (Table 3 and Table 4). These changes can affect not only their basic properties but also texture, content of bioactive compounds, antioxidant capacity or antibacterial efficiency.

## 6. Prophylactic and Therapeutic Properties of Plain Yoghurts

### 6.1. Source of Bioactive Proteins

Yoghurts are classified as functional food with prophylactic and therapeutic properties, inter alia, due to the presence of whey proteins. Many years of research, both medical and nutritional, have revealed that individual fractions show the ability to improve health and support people struggling with civilization diseases, including obesity, cancer or neurodegenerative diseases [118,119]. Consuming yoghurt with a daily diet allows one to control the appetite by stimulating the secretion of gastrointestinal hormones. Whey proteins contained in fermented milk products are dynamically digested, thanks to which they are able to quickly induce the feeling of satiety, increasing the concentration of amino acids, blood glucose and thermogenesis [120]. Bioactive peptides present in fermented foods and derived from the amino acids provided by milk proteins are also closely related to health. They mainly show hypocholesterolemic, antiinflammatory, as well as immunomodulating and antioxidant effects. These properties have contributed to their use in fortifying fermented products, positively affecting bones, adipose tissue and the gastrointestinal system, including the gastrointestinal microbiota [121,122].

### 6.2. Source of Vitamins

Yoghurt is a source of essential nutrients such as B vitamins, vitamin D_3_, phosphorus and calcium. In the context of cardiological diseases, calcium has a positive effect on the regulation of the blood lipid profile thanks to the increased ability to excrete fat in the stool [123]. Vatanparast et al. [124] demonstrated in their research that diets enriched with yoghurt were richer in macro- and microelements than ones that did not include this fermented product.

### 6.3. Other Health-Promoting Properties of Yoghurt

The consumption of fermented dairy products has become a better and more advantageous alternative to proinflammatory markers than unfermented dairy products, meat products or raw milk [125]. According to the available literature, the consumption of low-fat yoghurt is associated with a reduction in the development of type 2 diabetes [126]. Similar results were also reported by Brouwer-Brolsma et al. [127] and Trichia et al. [128] who demonstrated a relationship between the consumption of fat-free dairy products and pre-diabetes. Currently, there are many literature reports mentioning the health benefits of yoghurt consumption. The addition of yoghurt in the diet can greatly facilitate the consumption and absorption of nutrients and the maintenance of metabolic well-being. This allows you to maintain a levelled energy balance, thus having a positive effect on weight control [89,129,130,131,132]. D’Addezio et al. [133] showed that yoghurt eaters spend more time being physically active and show a greater interest in healthy food than ones who do not eat yoghurt. Yoghurts have a significant impact on lifestyle, and, consequently, the diet of various age groups. In one observational study conducted among the Italian population, Mistura et al. [134] indicated that people who consumed even a small amount of yoghurt every day had significantly higher diet quality parameters according to the PANDiet index when compared to people who did not consume milk or dairy products. Consumers who ate yoghurt had a higher intake of carbohydrates, total fat, PUFA, dietary fibre, B vitamins, calcium, zinc and iron. Therefore, the daily consumption of yoghurt may translate into overcoming mineral and vitamin deficiencies and improving the quality-of-life parameters. As a result of one of the greatest pandemics of the last millennium, fermented milk products have established themselves as functional foods. Thanks to the presence of probiotic bacteria, yoghurts help to maintain a healthy host microbiome and high resistance to pathogens from the external environment [135,136].

Example studies on the impact of eating yoghurt on the body in the case of the most common civilization diseases are presented in Table 6.

The therapeutic and preventive properties resulting from regular consumption of the fermented dairy product, yoghurt, are presented graphically in Figure 3.

In recent years, the probiotic supplementation of yoghurt has also attracted a lot of interest due to its increasingly suggested potential health benefits. This mainly concerns bacterial strains of the genus *Lactobacillus* and *Bifidobacterium* that are commonly used in the production of yoghurt [157]. Probiotic bacteria are living microorganisms that, when properly administered, are able to provide health benefits to the host. The viability of probiotics is influenced by many factors, including strain diversity, lactic acid build-up, the linkage relationship with starter cultures and product storage conditions [158]. Probiotic strains contribute to strengthening the intestinal microbiota, thanks to their ability to colonize the digestive tract. They stimulate immune functions, increase the bioavailability of nutrients, improve the blood lipid profile or reduce inflammation in the body inducing anticarcinogenic effects [159,160,161]. Ramachandran and Varghese [155] demonstrated that yoghurt containing probiotic bacteria can be a cheap and safe dietary supplement capable of alleviating the symptoms of incurable bowel disease, more specifically irritable bowel syndrome (IBS). By contrast, Li et al. [162] proved that the use of probiotic bacteria in the diet, specifically strains from the *Lactobacillus* group obtained from yoghurt, has a positive effect on the improvement in the quality and frequency of bowel movements in people struggling with a functional digestive disorder. Similar conclusions regarding the benefits of using probiotics were reached by Mirghafourvand et al. [163] who conducted a study among pregnant women struggling with constipation. They showed that consuming yoghurt enriched with *Bifidobacterium* and *Lactobacillus* 4.8 × 10^10^ cfu for a period of 4 weeks contributed to an increase in the number of bowel movements, including the improved colour and consistency of stools.

Moreover, Maghaddam et al. [164] investigated the benefits of using bacterial cultures to reduce bisphenol A in the digestive tract. They showed that the presence of *Lactobacillus acidophilus* and *Lactiplantibacillus plantarum* strains in yoghurt significantly decreased the concentration of bisphenol A by 43.44% for *Lactobacillus acidophilus* and 82.8% for *Lactiplantibacillus plantarum* during storage. Thus, they demonstrated that probiotic bacteria could capture toxins and prevent their absorption by the body.

Numerous studies have also shown the beneficial effect of combining LAB with fruit, vegetables and other plant substances classified as prebiotics, e.g., in the form of inulin or fructo-oligosaccharides. Enriching yoghurts with plant additives contributes to the fact that they become a synbiotic food, in which the synergistic interaction of probiotics and prebiotics helps to fight many diseases, including obesity and metabolic disorders [129,165,166,167,168]. In recent years, interest in probiotic yoghurts has also increased among pregnant women [169,170]. Ribeiro et al. [171] assessed the effect of using Jerusalem artichoke flour (*Helianthus tuberosus* L.) as a prebiotic yoghurt enrichment. They showed that fortification with Jerusalem artichoke flour had a bifidogenic function, and in combination with lactic acid bacteria, it gave yoghurt a health-promoting character, reducing the level of cholesterol and glucose in the blood. Moreover, the amount of inulin (the best-known prebiotic) in yoghurt increased significantly, and the product itself was characterized by an increased amount of dietary fibre. The higher the percentage of flour added, the more protein, fat and carbohydrates the yoghurt contained. The combination of probiotics and prebiotics, otherwise known as symbiotics, in yoghurts is desirable because it contributes to the prevention of many diseases [172,173]. Moreover, Bărboi et al. [174] demonstrated that inulin changes the composition of the gastrointestinal microflora by regulating the passage of the large intestine among people with intestinal disorders. IBS is a dysfunction caused by dysbiosis of the intestinal microflora that affects people of all ages. The hallmark of this disorder is recurring abdominal pain combined with a lack of regular bowel movements. Numerous studies have also been conducted to assess the effects of yoghurt consumption as a tool in the fight against metabolic syndrome (MetS), including obesity, cardiovascular disease and diabetes. MetS is currently an increasingly common health problem worldwide [175,176,177,178,179,180]. Both natural, plain and yoghurts with plant additives are beneficial for the host. Due to plant additives, fermented milk products increase their antioxidant capacity and become a synbiotic food, which is very desirable nowadays.

## 7. Prophylactic and Therapeutic Properties of Yoghurts with Plant Additives

Yoghurts can also be classified as novel foods, as they can contain exotic plants such as pomegranate fruit or baobab fruit pulp [181,182]. The enrichment of fermented milk products with foods such as fruits, vegetables, cereals, herbs or flowers is a part of the innovation of the dairy industry [183,184,185,186].

Fruits are an excellent source of antioxidants and dietary fibre. They are among the plant raw materials most frequently added to fermented dairy products because their combination with yoghurt provides synbiotic properties, increasing the stimulation of human microbiota [187]. A wide variety of fruits improve the quality of commercial products and provide health benefits to those who eat them. The greatest advantage of the presence of fruit in yoghurts is the increased antioxidant activity resulting from the introduction of polyphenols. Liu and Lv [188] showed that the addition of berry pulp helped in promoting the growth of *Lactiplantibacillus plantarum* and *Streptococcus thermophilus* bacteria and significantly enhanced the antioxidant effect of yoghurt. Maqsood et al. [189] confirmed in their research that the addition of 10% date syrup increased the antioxidant value of yoghurt, thereby improving the nutritional value of the fermented milk product. In turn, Kennas and Amellal-Chibane [190] enriched yoghurt with honey and pomegranate peel powder. They found that a small addition of these two ingredients had positive sensory effects and did not disturb the fermentation process. The fortified product was also distinguished by a higher content of phenols and flavonoids, and higher antioxidant activity.

Other functional additives that can be found in yoghurts include vegetables, flowers, fruits and herbs. Yildiz and Ozcan [35] assessed the effect of adding vegetable puree (pumpkin, carrots, green peas and zucchini) on the content of ascorbic acid and total phenolics. The addition of individual vegetables (pumpkin, carrots, green peas and zucchini) in the form of a puree increased the amount of ascorbic acid (amounting to, respectively: 8.87, 3.54, 5.31, 7.08 mg/kg) and total phenolics (75.17, 47.28, 52.37, 47.46/mg GAE/kg, respectively) versus yoghurt without the addition of vegetables. The presence of fruit, vegetables or flowers in yoghurts increases the total content of polyphenols, which translates into an improvement in their antioxidant capacity [191]. Bioactive compounds contained in plant raw materials interact with milk proteins, forming complexes that increase the content of phenols in fermented milk products [192]. Raikos et al. [193] demonstrated that the addition of Salal (*Gaultheria shallon*), blueberries and currant pomace (*Ribes nigrum*) increased the level of phenol in yoghurt drinks as compared to control yoghurt. This increase may be correlated with the capacity of tyrosine to release amino acids during milk protein proteolysis [194]. Similar results, indicating an increase in phenol content in fermented milk products, were obtained by Nguyen and Hwang [111] who added chokeberry juice to yoghurt. Amirdivani and Baba [195] enriched yoghurts with extracts of peppermint (*Mentha piperita*), fennel (*Foeniculum vulgare*) and basil (*Ocimum basilicum*) and reported that thanks to the presence of bioactive peptides, the herbal additives increased the products’ proteolytic and antioxidant activity. All herbal yoghurts showed higher anti-ACE (angiotensin-converting enzyme) activity than plain yoghurt at corresponding storage periods. Gaglio et al. [196] assessed the effect of adding saffron to yoghurt. They showed that saffron was an acceptable additive that increased the antioxidant properties of the dairy product and extended the viability of yoghurt starter cultures, i.e., *Streptococcus thermophilus* and *Lactobacillus delbrueckii* subsp. *bulgaricus*, despite the long storage time (60 days). In addition, with the passage of time, the antioxidant capacity increased, which was also an advantage resulting from the addition of this spice. Hasneen et al. [40] showed that the addition of 10% water extracts of turmeric (*Curcuma longa*), sage (*Salvia officinalis*) and marjoram (*Origanum majorana*) increased the total content of phenols and total flavonoids. In the case of phenols it was at the level of 3.14–4.32 mg/g for the yoghurts with turmeric, sage and marjoram (in control yoghurt—3.09 mg/g), and for the total amount of flavonoids, determined by the level of quercetin, it ranged from 1.39 mg/g (turmeric) to 1.48 mg/g (marjoram), as compared to the control yoghurts—1.12 mg/g. The free radical scavenging capacity (RAS) was 26.72–29.56% for the yoghurts with additives and 20.56% for the control yoghurt. Other studies indicate that it is also possible to use green and black tea as yoghurt additives. Tea is an excellent antioxidant, increasing the total polyphenol content compared to the control. The total phenolics content in the control yoghurt was 118 g GAE/mL, while in yoghurt containing green, protein and black tea it was 479.48 g GAE/mL, 366.89 g GAE/mL and 27,837 g GAE/mL, respectively [194]. Moreover, the content of phenols (TPC) in the product containing the additive increased by as much as 11 times, i.e., from 11.64 µg GAR/mml in the control yoghurt to 113.58 µg GAR/mml in yoghurt containing 1% of the additive. This evidenced the enormous antioxidant potential of Chinese green tea extract. Similar results, indicating an increase in phenol content in fermented milk products, were obtained by Nguyen and Hwang [111] who added chokeberry juice to yoghurt, and by Dönmez et al. [41], who fortified yoghurt with powdered green coffee and tea. On the other hand, Caleja et al. [197] considered the differences between a natural and a synthetic additive in the form of potassium sorbate in the context of shaping the antioxidant capacity of yoghurts. They evaluated decoctions of *Foeniculum vulgare Mill.* (fennel) and *Matricaria recutita* L. (chamomile) and demonstrated that the fortification of yoghurts with these natural additives resulted in higher antioxidant activity of the yoghurt and was more efficient than the synthetic alternative. On the other hand, Van Nieuwenhove et al. [34] assessed yoghurt fortification with jacaranda seeds (*Jacaranda mimosifolia* D. Don). They showed that the antioxidant activity, expressed as DPPH (I%), decreased from 44.05 to 41.65 during 28 days of storage. In the control yoghurt the anti-free radical activity (I%) decreased from 30.1 to 29.25 (I%) during storage. Moreover, Shori [198] assessed the antioxidant properties of polyphenol extract form nutmeg, black pepper and white pepper seeds. He indicated that plant yoghurts had significantly higher TPC (respectively 29.09 ± 1.0, 34.41 ± 1.9 and 27.08 ± 1.4 µg GAE/mL) than the control (22.69 ± 0.8 µg GAE/mL). After 21 days of storage, the highest TPC was for black pepper (46.53 µg GAE/mL), and for nutmeg it was—37.45 µg GAE/mL and white pepper—34.96 µg GAE/mL, while the control yoghurt had TPC at the level of 20.587 µg GAE/mL.

The addition of nuts and oilseeds increased the level of unsaturated fatty acids in the product. Baba et al. [199] evaluated the effect of adding walnut oil and linseed to yoghurt with guar gum. They showed that their addition increased the antioxidant activity and oxidative stability of the yoghurt. Fortification with walnut oil resulted in higher quality parameters and a higher content of essential fatty acids (MUFA and PUFA) than in the case of linseed oil.

Grains of cereals, e.g., wheat, rye, oat, millet, buckwheat or rice, are a frequent addition to fruit or plain, natural yoghurts. As part of the innovation in the dairy production, however, more unusual additives have been proposed such as mushrooms, seed mucilage and peels—Table 3.

It should also be mentioned that the functionality of yoghurts can be enhanced by adding other synthetic substances, especially vitamins or minerals. They can facilitate increased consumer acceptance.

## 8. Conclusions

The variety of plant additives (fruits, vegetables, cereals, nuts, seeds, oils, plant or herbal extracts, fruit or vegetable fibre and waste from fruit processing) proposed in the research is huge and very diverse in different regions of the world. Therefore, the plant additives affect the characteristics of yoghurts in different ways. Plant additives used in the production of yoghurts have a direct impact on the acceptability of the final product. In general, yoghurts enriched with plant additives are more valuable, especially in terms of the content of health-promoting compounds such as fibre, phenolic compounds, vitamins, fatty acids and minerals. These bioactive compounds may contribute to protecting the body against modern civilization diseases. Plant-derived antioxidants delay the oxidation of dairy products, thus extending their shelf life. Studies carried out in recent years have shown that fermented milk product, yoghurt, can serve as prophylactic and therapeutic food in the context of many diseases. In addition, no correlation between the yoghurt consumption and mortality due to any diseases has been observed. However, the use of additives may impair the microbiological, texture and sensory properties of the yoghurts. Therefore, it is important to conduct comprehensive research before launching a new product on the market.

## Figures and Tables

**Figure 1 foods-12-01275-f001:**
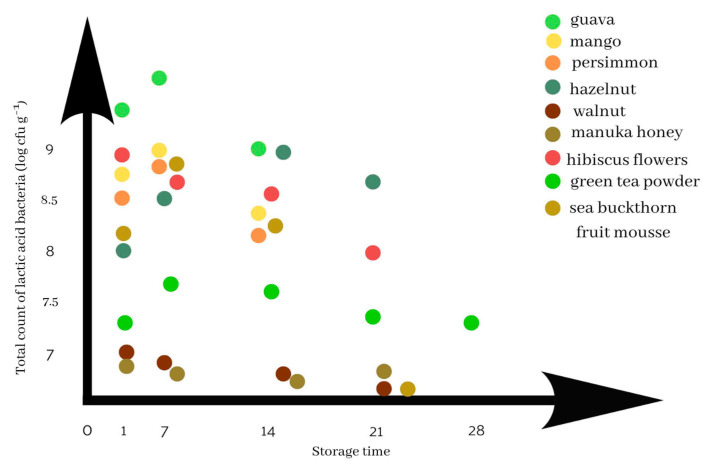
Microbiological changes in the yoghurts with plant additives during storage.

**Figure 2 foods-12-01275-f002:**
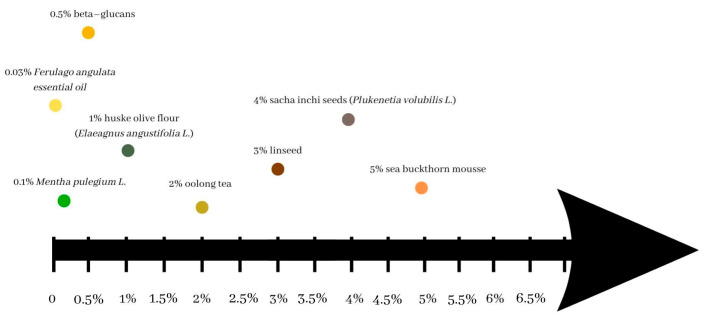
Maximum % of plant additive and positive overall acceptability (taste, odour and appearance) in the yoghurts.

**Figure 3 foods-12-01275-f003:**
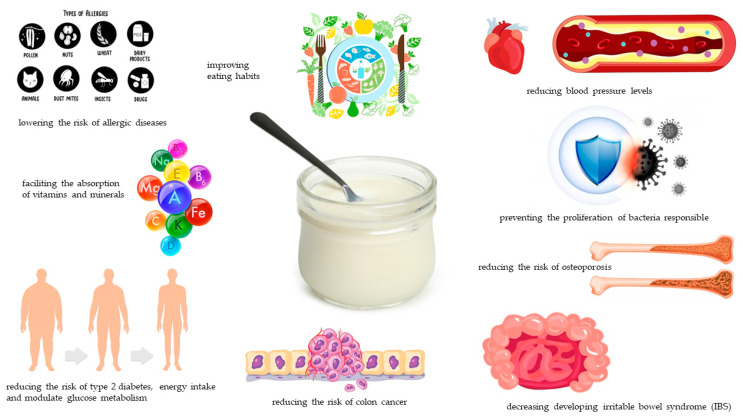
Therapeutic and preventing properties resulting from regular consumption of yoghurts.

**Table 1 foods-12-01275-t001:** Scale of milk and yoghurt production globally, in Europe and in Poland [4,5,6].

Production	Milk (MM t) *	Yoghurt (MM t) *	Percentage (%)
World	132,601.202	230.801	0.17
Europe	229	8.20	3.58
Poland	1.447	0.73	5.04

* MM t—million tonnes.

**Table 2 foods-12-01275-t002:** Comparison of the content of basic ingredients and physical properties of natural yoghurt made from milk obtained from various animal species [11,12,13,14,15,16,17,18,19,20,21].

Composition	Animal Species				
	Cow	Sheep	Goat	Buffalo	Camel
Basic Nutritional Value					
Total protein (%)	3.2–4.6	5.1–9.5	3.9–7.9	4.2–4.7	3.2–3.6
Casein (%)	2.5–2.7	3.9–7.6	2.9–6.3	3.2–3.7	2.2–2.8
Fat 9%)	3.5–5.3	7.0–9.3	5.3–6.3	5.5–6.4	3.1–4.4
Lactose (%)	4.0–4.7	4.7–5.0	3.3–4.2	4.8–5.2	3.4–6.9
Lactic acid (%)	0.7–1.1	0.7–0.8	0.8–0.9	0.8–0.9	0.8–1.3
Ash (%)	0.7–0.8	0.9–1.5	0.7–0.8	0.7–0.8	0.8–1.1
Minerals					
Calcium (mg/100 g)	98–245	295–330	84–160	142–197	72–130
Phosphorus (mg/100 g)	88–170	180–268	89–140	125–200	50–72
Magnesium (mg/100 g)	10–13	19–22	8–15	10–16	9–13
Potassium (mg/100 g)	26–42	100–223	127–162	58–112	120–160
Physical features					
Active acidity (pH value)	4.4–5.2	4.4–4.7	4.2–4.5	4.3–4.5	4.3–5.6
Syneresis (%)	22–56	5–28	7–47	10–12	45–58
Moisture (%)	75–84	73–80	72–80	82–86	83–88

**Table 3 foods-12-01275-t003:** Selected studies on the yoghurts with the plant-based additives.

Additive Used	Evaluated Parameters	Experimental Approach—Factors Included	Results	References
Fruits				
Pineapple—powder peel	– physicochemical (pH value, water holding capacity), – textural, – rheological (texture, hardness), – sensory (colour), – microstructural	– storage time (28 days), – storage temperature (4 °C), – strains (*Lactobacillus acidophilus*, *Lacticaseibacillus casei*, *Lacticaseibacillus paracasei* ssp. *paracasei*)	Syneresis ↑ pH value ↓ firmness ↑ viscosity ↑ colour intensity ↑ probiotic cultures ↓	[51]
Grape—extract from grapes from *Vitis vinifera*	– microbiological (microbial growth), – physicochemical (syneresis, pH value), – rheological, – sensory	– storage time (12 days), – storage temperature (4 °C), – strains (*Lactobacillus delbruecckii* ssp. *bulgaricus* and *Streptococcus salivarius* ssp. *thermophilus*, *Lactobacillus acidophilus* and *Bifidobacterium bifidum*)	microbial growth ↓ pH value ↓ syneresis ↑ strength ↓ sensory properties ↓	[52]
Cantaloupe (*C. melon*)—powder and puree	– physicochemical (composition, moisture, ash, lipids, proteins, amino acids, pH value, water holding capacity—WHC, antioxidant activity), – microbiological, – rheological (viscoelasticity, viscosity)	– storage time (28 days), – storage temperature (4 °C), – strains (*Lactobacillus delbruecckii* ssp. *bulgaricus* and *Streptococcus salivarius* ssp. *thermophilus*, *Lactobacillus acidophilus* and *Bifidobacterium bifidum*)	pH value ↓ antioxidant activity ↓ WHC ↓ lactic flora ↓ quality of yoghurts ↑	[53]
Mango—juice	– physicochemical (titrated acidity—TA, pH), – microbiological, – textures, – sensory	– storage time (28 days), – storage temperature (4 °C), – strain (*Lactobacillus acidophilus*)	number of bacteria ↓ pH value ↓ viscosity ↑ TA ↑ colour value ↑	[54]
Lemon, grape, pawpaw, orange—juice, synthetic pineapple—flavourant	– physicochemical qualities (pH value, lactic acid, moisture, titratable acidity, syneresis, water holding capacity—WHC), – rheological (viscosity), – peroxide value, – vitamin C, – cholesterol value, – lactose, – free fatty acids—FFA, – sensory (colour, flavour, mouth feel, consistency, overall acceptability)	– fresh cow milk, – commercial freeze-dried starter (*Streptococcus thermophilus*, *Lactobacillus bulgaricus*), – storage time (14 days)	WHC ↑ viscosity ↓ syneresis ↓ moisture ↑ ash ↓ fat ↓ protein ↓ carbohydrates ↑ pH value ↑↓ lactic acid ↑ vitamin C ↓ FFA ↓ lactose ↑ cholesterol ↓ peroxide value ↑	[55]
Date—juice	– physicochemical qualities (pH value, titratable acidity, syneresis, total lactic acid bacteria)	– whole cow milk, – strains (*Lactobacillus acidophilus*. *Streptococcus thermophilus*, *Bifidobacterium longum*), – storage time (21 days), – storage temperature (4 °C)	pH value ↓ total acidity ↑ syneresis ↓ total lactic acid bacteria ↓	[56]
Vegetables				
Pumpkin, carrots, green peas, zucchini—puree	– physicochemical (acidity—pH value, lactic acid, syneresis, colour instrumentally, total solids, ash, dietary fibre, total antioxidant capacity, ascorbic acid, total phenolic content, total carotenoid), – texture (firmness, cohesiveness, consistency, viscosity index), – sensory (appearance, texture, aroma, taste, colour, flavour, sensorial acidity, general aspect), – microbial (*Streptococcus thermophilus* and *Lactobacillus delbrueckii* subsp. *bulgaricus*)	– fresh raw cow milk, – starter culture (*Streptococcus thermophilus* and *Lactobacillus delbrueckii* subsp. *bulgaricus*), – puree addition (10 g/100 g), – time of storage (28 days)	pH value ↓ number of bacteria ↓ titratable acidity ↓ syneresis ↓ total solids ↓ ash ↓ dietary fibre ↑ total carotenoid ↑ total antioxidant capacity ↑ ascorbic acid ↑ total phenolic content ↑	[35]
Chicory—inulin extracted from chicory roots (*Cichorium intybus* L.)	– physicochemical (pH value, titratable acidity, water holding capacity—WHC), – texture (hardness, adhesiveness, cohesiveness, springiness, gumminess and chewiness), – sensory (appearance, texture, aroma, taste, colour, overall acceptability), – microbial (*Streptococcus thermophilus*, *Lactobacillus acidophilus* and *Bifidobacterium bifidum*), – microstructural characterization	– raw cow milk (3% fat), – probiotic starter culture (*Streptococcus thermophilus*, *Lactobacillus acidophilus* and *Bifidobacterium bifidum*), and yoghurt—starter culture (*Streptococcus thermophilus* and *Lactobacillus delbrueckii* subsp. *bulgaricus*), – low-fat synbiotic yoghurt, – inulin (0.5, 1, 2%), – time of storage (14 days)	microstructural properties ↑ rheological properties ↑ sensory properties ↓ pH value ↑ titratable acidity ↑ WHC ↓ microbial ↓	[57]
Potato peel flour	– physicochemical (moisture, ash, protein, total dietary fiber, syneresis, acid production, pH value, apparent viscosity), – total ethereal extract, – total soluble carbohydrates, – total polyphenol content, – antioxidant capacity (ABTS), – prebiotic activity score	– rehydrated skimmed milk (10% *w*/*v*), – starter culture (*Streptococcus thermophilus*, *Lactobacillus delbrueckii* subsp. *bulgaricus*), – storage time (21 days)	syneresis ↓ apparent viscosity ↑ titratable acidity ↑ pH value ↓ sensory acceptation ↓	[58]
Herbs and tea				
Green coffee powder and green tea powder	– physical (pH value, syneresis, colour instrumentally), – rheological behaviour	– pasteurized and homogenized – milk (3% protein, 3% fat) – starter culture (*Streptococcus thermophilus* and *Lactobacillus delbrueckii* subsp. *Bulgaricus*) – green coffee powder and green tea powder (0, 1, 2%) – time of storage (14 days)	syneresis ↓ pH value ↓ shear stress ↑ flow behaviour index ↓ colour—no significant differences	[59]
Basil gum from basil seeds	– physicochemical (pH value, total titration acidity, viscosity, colour instrumentally, syneresis, fat, moisture, total phenol, flavonoid, antioxidant activity), – sensory (appearance, colour, flavour, texture, sourness, overall acceptability),	– milk, reduced-fat milk and non-fat milk, – basil seed gum (0, 0.5, 1%)	total phenol content ↑ pH value ↓ titratable acidity ↑	[60]
Flowers				
Blue pea flower (*Clitoria ternatea* L.)	– physicochemical (2,2-diphenyl-1-picrylhidrazyl (DPPH) antioxidant levels), – sensory (colour)	– liquid skimmed milk, pasteurized milk, pasteurized milk added by skim powder, UHT milk, and UHT milk added by skim powder, – strains (*Lactobacillus bulgaricus*, *Streptococcus thermophiles*), – blue pea flower (10% (*v*/*v*) of the milk volume)	antioxidant activity ↑ colour ↑	[61]
Blue pea flower (*Clitoria ternatea*)	– antioxidant activity (DPPH antioxidant levels), – colour properties	– lactic acid bacteria’s fermentation (*Lactobacillus* bulgaricus, *Streptococcus thermophilus*), – 10% extract of blue pea flower and the addition of sucrose by 0%, 4%, 8% and 12% (*v*/*v*)	antioxidant activity ↑ colour ↑	[62]
*Hibiscus sabdariffa* L. flowers	– physicochemical (dry matter, ash, protein, pH, titratable acidity (anhydrous citric acid, %), water soluble dry matter, water activity, syneresis, antioxidant acidity, total phenolic content), – texture (viscosity), – sensory (appearance, texture, aroma, taste, colour, overall acceptability), – microbial (*Lactobacillus bulgaricus*, *Streptococcus thermophiles*, *Escherichia coli*, *Staphylococcus aureus*)	– whole cow milk (3% fat), – strains (*Lactobacillus bulgaricus*, *Streptococcus thermophiles*), – *Hibiscus sabdariffa* L. flowers (0, 15, 20%), – time of storage (21 days)	pH value ↓ titratable acidity ↑ total solids ↑↓ fat ↑↓ protein ↓ ash ↑ lactic acid ↓ viscosity ↑↓ syneresis ↑↓ serum separation ↓ macro minerals ↓ micro minerals ↓ antioxidant activity ↑ sensory properties ↓	[63]
Seeds				
Lyophilized tamarind (*Tamarindus indica* L.) seed kernel powder	– physicochemical (fat, total solids, acidity), – sensory (appearance, flavour, body and texture)	– cow milk, – strains (*Lactobacillus bulgaricus*, *Streptococcus thermophiles*), – lyophilized tamarind seed kernel powder (0, 0.1, 0.25, 0.5%)	pH value ↑ acidity↓ fat ↑↓ total solids ↑ specific gravity ↓ sensory properties ↓	[64]
Watermelon (*Citrullus lanatus*) seeds powder	– physicochemical (protein, fat, titratable acidity, total solid, ash), – sensory (colour, flavour, texture, taste, overall acceptability)	– fresh cow milk, – strains (*Lactobacillus bulgaricus*, *Streptococcus thermophiles*), – watermelon seeds powder (5, 7, 10%) – time of storage (10 days)	protein ↓ fat ↑ acidity ↑ total solids ↓ ash ↓ colour ↓↑ flavour ↓ texture ↓↑ taste ↓↑ overall ↓↑	[65]
Grape seed—extract	– physicochemical (total solids, protein, ash, pH value, water holding capacity—WHC), – total phenolic content—TPC, – texture (viscosity), – sensory properties (colour, flavour, texture, overall acceptability), – *in vitro* antioxidant, anti-bacterial, and anticancer activities	– buffalo skimmed milk, – grape seed extract—GSE (*Vitis vinifera*) (0, 0.1, 0.25, 0.5%)	total solids ↑ ash ↑ pH value ↑ WHC ↑ viscosity ↑ total phenolic content ↑ colour ↓ flavour ↓ texture ↓ overall acceptability ↓ *in vitro* antioxidant, anti- bacterial, and anticancer activities ↑	[66]
Nuts				
Hazelnut, walnut, almond, pistachio nuts	– physicochemical (total protein, pH, titratable acidity, syneresis, water holding capacity—WHC), – texture (firmness), – sensory (appearance, texture, aroma, taste, colour, overall acceptability), – folic acid, selenium, α- and γ-tocopherol, fatty acid content, total oil content	– fresh cow milk (3.6% fat), – strains (*Lactobacillus bulgaricus*, *Streptococcus thermophiles*), – hazelnut, walnut, almond, pistachio nuts (5%), – time of storage (21 days)	yoghurt bacteria ↓ pH value ↓ titratable acidity ↑ fat ↓ total protein ↓ syneresis ↓↑ WHC ↓ whey separation ↑ firmness ↑ folic acid ↓ selenium ↓ α- and γ-tocopherol values ↓↑ omega fatty acid composition ↓↑	[37]
Walnut (*Juglans regia* L.)	– physicochemical (dry matter, whey separation, total solids content, titratable acidity, syneresis, total phenolic content, total antioxidant activity) – sensory (appearance, consistency, taste–aroma, odour and general acceptability)	– cow milk (3% fat, 3.6% protein, 12.1% total solid content and 8 °SH acidity), – strains (*Lactobacillus bulgaricus*, *Streptococcus thermophiles*), – walnuts (0, 1, 2.5, 3.5, 5%) – time of storage (28 days)	titratable acidity ↑ pH value ↓ whey separation ↓ dry matter ↓↑ total phenolic content ↓↑ overall acceptability ↓↑ antioxidant activity ↓	[67]
Other				
Quince seed mucilage powder	– physicochemical (protein, fat, acidity—pH value, titratable acidity, syneresis, colour values), – texture (firmness, consistency, cohesiveness), – sensory (appearance, texture, aroma, taste, colour, flavour, sensorial acidity, general aspect), – microbial (*Streptococcus thermophilus* and *Lactobacillus delbrueckii* subsp. *bulgaricus*), – microstructure (SEM)	– raw cow milk (total solid 12.99%, milk fat 3.05%, titratable acidity 0.19% and pH 6.67), – strains (*Lactobacillus bulgaricus*, *Streptococcus thermophiles*), – lyophilised quince seed mucilage (0.05, 0.10, 0.15, 0.2%), – time of storage (21 days)	syneresis ↓ total solids ↑ fat ↓ protein ↓↑ ash ↓↑ viable counts ↓ pH value ↓ titratable acidity ↑ apparent viscosity ↓↑ colour—no significant changes firmness ↓↑ consistency ↓ cohesiveness ↓↑	[68]
Sour orange, sweet orange, lemon peels	– physicochemical (titratable acidity, moisture, total phenolic content), – sensory (overall acceptability (9-point hedonic scale)), – microbial (viability of starter culture, antioxidant activity, antibacterial activity)	– fresh cow milk: fat 3.2%, protein 3.3%, total solids 12.4%, – strains (*Lactobacillus bulgaricus*), – sour orange, sweet orange, lemon peels (0.5%), – time of storage (28 days)	titratable acidity ↑ moisture ↓ antioxidant acidity ↑ antibacterial efficiency ↓↑ viable counts ↓↑	[69]
Oyster mushroom (*Pleurotus ostreatus*)—powder	– total lactic acids, – acidity (pH value), – lactic acid bacteria (LAB) count, – organoleptic properties (colour, taste, flavour and texture)	– fermentation yoghurt (fresh cow milk, 10% skim milk powder, 10% sugar), – *Streptococcus*, *Lactobacillus acidophilus*, *Lactobacillus bulgaris*, – mushroom powder concentrations (0, 0.5, 1, 1.5%)	total lacid acid ↑ pH value ↓ probiotic viability ↑ organoleptic properties ↑	[70]
Mushroom (*Pleurotus plumonarius*)—powder	– total solids, fat, total protein contents, titratable acidity and dietary fiber, pH value, total phenolic content—TPC), – RSA—radical scavenging activity, – total bacterial count, – syneresis, – viscosity, – sensory properties (body, texture, appearance)	– low-fat buffalo’s milk (fat 1%), – ABT-5 culture containing *Streptococcus thermophilus*, *Lactobacillus acidophilus* and *Bifidobacterium bifidum*, – mushroom powder (1, 2, 3%)	total solids ↑ fat ↑ protein ↑ dietary fiber ↑ acidity↑ pH ↓ total phenolic content ↓ RSA ↓ syneresis ↑ viscosity ↑ total bacterial count ↓	[71]

↓ The decrease in the parameter value in yoghurts with plant additives. ↑ The increase in the parameter value in yoghurts with plant additives.

**Table 4 foods-12-01275-t004:** Summary of the effect of plant additives on the physicochemical, microbiological and sensory characteristics of yoghurts (own elaboration).

Parameters	Plant Additives				
	Fruits	Vegetables	Herbs and Tea	Seeds and Nuts	Flowers
pH value	↓	↓	↓↑	↑	↓
Syneresis	↑	↑	↓↑	↑	↓↑
Colour	↓	↓	↑	↑	↓
Total phenolic content—TPC	↑	↑	↑	↑	↑
Bacteria content	↓	↑	↓	↓	↓
Antioxidant activity	↑	↑	↑	↑	↑
Protein content	↑	↓	↑	↓	↓
Firmness	↑	↑	↑	↑	↑
Titratable acidity	↑	↑		↑	↑
Apparent viscosity values	↑	↑	↓↑	↑	↑
Water holding capacity—WHC		↓	↑	↓↑	↓
Sensory properties		↓			↓

↓ The decrease in the parameter value in the yoghurts with plant additive. ↑ The increase in the parameter value in the yoghurts with plant additive.

**Table 5 foods-12-01275-t005:** Selected studies on the plain (natural) yoghurts.

Additive Used	Evaluated Parameters	Experimental Approach—Factors Included	Results	References
Natural yoghurt on the basis of typical cultures	– physicochemical (pH value, titratable acidity), – colour, – the number of bacteria, – sensory (appearance, texture, taste, smell, desirability)	– storage time (21 days), – storage temperature (8 °C and 2 °C), – strains (*Lactobacillus delbruecckii* ssp. *bulgaricus* and *Streptococcus salivarius* ssp. *thermophilus*)	acidity ↑ pH value ↓ number of bacteria ↓ sensory parameters ↓	[115]
Natural yoghurt with added cultures of *Propionibacterium shermanii* subsp. *freudenreichii*	– physicochemical (protein content, total solids, titratable acidity, pH value), – lactose content, – texture (hardness), – numbers of bacteria, – sensory (colour, appearance, taste, aroma, texture, overall acceptability	– storage time (21 days), – storage temperature (4 °C), – strains (*Lactobacillus acidophilus*, *Propionibacterium shermanii* subsp. *freudenreichii*)	pH value ↑ acidity ↓ probiotic bacteria ↑ sensorial properties ↓	[116]
Natural yoghurt on the basis of typical cultures	– physicochemical (pH value, acidity, water holding capacity—WHC), – rheological (texture), – microbiological (survival and antifungal activity, lactic acid bacteria—LAB), – sensory	– storage time (30 days), – storage temperature (4 °C), – strains (*Limosilactobacillus reuteri*, *Lactobacillus helveticus*, *Lactobacillus acidophilus*)	LAB ↑ pH value ↓ titratable acidity ↓ WHC ↓ sensorial properties ↑	[117]

↓ The decrease in the parameter value in the plain yoghurts. ↑ The increase in the parameter value in the plain yoghurts.

**Table 6 foods-12-01275-t006:** Exemplary studies on the impact of eating yoghurt on the body in the case of the most common civilization diseases.

Disease/Dysfunction	Research with Yoghurt Health Effect	Literature Reference
Metabolic syndrome (MetS)/Syndrome X	Consumption of reduced-fat yoghurt was closely associated with lower fasting glucose, blood pressure and lipid parameters. Compared to those consuming the smallest amount of whole dairy products (up to 270 g/day), participants consuming dairy products above 450 g/day were distinguished by higher physical activity, lower blood triglycerides concentration and higher high-density lipoprotein concentration. In addition, they consumed fewer calories, and less red meat, grains, nuts and alcohol with their diets.	[137,138]
Cancer	Consumption of yoghurts showed a reduction in the odds ratio (OR) of the occurrence of colorectal cancer (OR = 0.88, 95% CI = 0.84–0.93), bladder (OR = 0.79, 95% CI = 0.68–0.91) and oesophageal cancer (OR = 0.64, 95% CI = 0.54–0.77).	[139]
Insulin resistance	People consuming 220 g of yoghurt daily for 24 weeks showed a reduction in the HOMA–IR (Homeostatic Model Assessment of Insulin Resistance) index from 3.78 to 2.8. It also lowered: – fasting insulin level (from 15.33 to 12.55 mU/L), – the number of triglycerides in the blood (–0.19 mmol/L), – the amount of fibroblast growth factor 21 (−57.76 pg/mL), – the level of lipopolysaccharides (–0.31 EU/mL), – human serine protease inhibitor specific for visceral adipose tissue (−0.05 ng/mL).	[140,141]
Diabetes	Consumption of yoghurt may reduce the risk of type 2 diabetes in older healthy adults and adults at high cardiovascular risk. Yoghurt consumption can modulate glucose metabolism and gives a feeling of satiety (due to the high-quality amino acids), which reduces energy intake and regulates blood glucose levels.	[142,143,144]
Hypertension	Consumption of two or more yoghurts for a month reduces blood pressure levels. In addition, people consuming yoghurt showed greater physical activity, higher efficiency DASH diet, higher amounts of dietary fibre, vegetables and fruits, and lower consumption of alcohol, red meat and meat products.	[145,146,147]
Osteoporosis	Consumption of yoghurt by women reduced the risk of osteoporosis by 31% and men by 52%. Consumption of yoghurt in the elderly may help maintain bone health. People who consumed one or more daily servings of yoghurt had a 3.1% higher total hip density than people who do not consume fermented milk products. Due to the content of key minerals and vitamins (calcium, vitamin D, vitamin B2 and B12), yoghurt has protective properties for changes in the bone and joint system.	[148]
Caries	Natural yoghurt without added sugar, due to the presence of lactic acid, contributed to lowering the pH in the oral cavity, thus inhibiting the multiplication of pathogenic microorganisms. In addition, the ongoing fermentation process contributed to the reduction in lactose levels, which reduces the formation of caries in the oral cavity.	[149]
Food allergies	Among the 1166 children surveyed whose parents declared responses at the age of 5, it was shown that their intake of fermented milk products in infancy lowered the risk of allergic diseases. Moreover, among 5 year olds the concentration of specific IgE indicated that only 8.8% were allergic to food allergens, compared to the total allergy, which was less than 58%.	[150,151,152]
Irritable bowel syndrome (IBS)	Research has shown that regular yoghurt consumption is correlated with a lower likelihood of developing irritable bowel syndrome (IBS). Moreover, in one of the studies, it was shown that people (150) who consumed about 700 g of yoghurt per day achieved disease remission within 180 days (91% of respondents) and achieved complete recovery within 300 days (about 97% of respondents). Due to the presence of lactic acid bacteria (LAB), yoghurt belongs to the food that is a natural probiotic, having a health-promoting effect on the mycophore of the gastrointestinal tract—therapeutically in the case of IBS.	[153,154,155]
Coronavirus 2 (SARS-CoV-2)	Studies on the impact of food and Covid-19 indicate that enriching the diet with fermented milk products, i.e., yoghurt, prevents the proliferation of bacteria responsible for the development of respiratory infections. Due to the rich nutritional matrix offered by yoghurt, it improves the functioning of the digestive tract, thus strengthening the overall immunity of the body.	[156]

## Data Availability

No new data were created or analysed in this study. Data sharing is not applicable to this article.

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
