# Peer review of "Shaping the Physicochemical, Functional, Microbiological and Sensory Properties of Yoghurts Using Plant Additives"

_foods, 2023, doi:10.3390/foods12061275_

Round 1
Reviewer 1 Report
Dear Editors and authors,
The review (Shaping the physicochemical, functional, microbiological and 2 sensory properties of yoghurts by the plant additives) needs some corrects and modifications.
1- The tittle of Table 1 correct to Scale of milk and yoghurt production in the world, Europe and Poland.
2-Table 1 , The standard unit of column 2 and 3 is unclear for readers.
3-Table 2, The results are not consistent with the chemical composition of the yoghurt, especially the percentage of moisture in the yoghurt produced from goats. The table must be reviewed.
4- Under table 2, The symbol (nd) is not found in table. delete this symbol.
5- The term (Animal species) placed in the middle of the first row in the table 2.
6-Table 4, The arrows in the table have no meaning and have not been explained or clarified. An explanation or clarification must be added below the tables to show its meaning.
7-The manuscript contains Table 4 mentioned twice, the table numbers should be corrected.
8-In 2020, a new name was used for lactic acid bacteria. Composers should use these names, such as Lactobacillus plantarum correct to Lactiplantibacillus plantarum, See line 453, 456 table 6 and table 7. The modern nomenclature was adopted throughout the manuscript.
9-Table 7, The arrows in the table have no meaning and have not been explained or clarified. An explanation or clarification must be added below the tables to show its meaning.
10-The authors used many scientific references in order to increase the modernity of the references. I suggest to add
Al-Sahlany, S. T. G., Khassaf, W. H., Niamah, A. K., & Al-Manhel, A. J. Date juice addition to bio-yogurt: The effects on physicochemical and microbiological properties during storage, as well as blood parameters in vivo.
Part, N., Kazantseva, J., Rosenvald, S., Kallastu, A., Vaikma, H., Kriščiunaite, T., ... & Viiard, E. (2023). Microbiological, chemical, and sensorial characterisation of commercially available plant-based yoghurt alternatives. Future Foods, 7, 100212.
Tami, S. H., Aly, E., Darwish, A. A., & Mohamed, E. S. (2022). Buffalo stirred yoghurt fortified with grape seed extract: New insights into its functional properties. Food Bioscience, 47, 101752.
Author Response
Dear Reviewer 1,
Thank you for giving us the opportunity to submit a revised version of our manuscript titled “Shaping the physicochemical, functional, microbiological and sensory properties of yoghurts using plant additives” to Foods.
We appreciate the time and effort that you have dedicated to providing your valuable feedback on our manuscript. We are grateful for your comments on our paper. We have made corrections in the text in accordance with the suggestions.
We would also like to emphasize that in the terms of English language, the entire text has been proofread by a sworn translator.
Reviewer: 1- The tittle of Table 1 correct to Scale of milk and yoghurt production in the world, Europe and Poland.
Authors: Thank you for this suggestion. It has been corrected in the text (L 44).
Reviewer: 2-Table 1 , The standard unit of column 2 and 3 is unclear for readers.
Authors: Thank you for the suggestion. The standard unit has been corrected and an explanation has been added under the table (L 45).
Reviewer: 3-Table 2, The results are not consistent with the chemical composition of the yoghurt, especially the percentage of moisture in the yoghurt produced from goats. The table must be reviewed.
Authors: Previously, the averaged values from various publications were given in the Table 2, which could be the reason that they did not sum up. In the current version, it has been corrected to content ranges (L 62, Table 2).
Reviewer: 4- Under table 2, The symbol (nd) is not found in table. delete this symbol.
Authors: Thank you for the comment. The symbol has been removed (L 63).
Reviewer: 5- The term (Animal species) placed in the middle of the first row in the table 2.
Authors: Thank you for your suggestion. The term has been placed in Table 2 (L 62).
Reviewer: 6-Table 4, The arrows in the table have no meaning and have not been explained or clarified. An explanation or clarification must be added below the tables to show its meaning.
Authors: Thank you for the comment. The arrows have been explained under the Tables 3, 4 and 5 (L 193-194, 372-373, 380-381).
Reviewer: 7-The manuscript contains Table 4 mentioned twice, the table numbers should be corrected.
Authors: Thank you for the comment. It has been corrected.
Reviewer: 8-In 2020, a new name was used for lactic acid bacteria. Composers should use these names, such as Lactobacillus plantarum correct to Lactiplantibacillus plantarum, See line 453, 456 table 6 and table 7. The modern nomenclature was adopted throughout the manuscript.
Authors: Thank you for the suggestion. It has been corrected in the text of manuscript (L 464, 466, 506) and in Tables 3 and 5.
Reviewer: 9-Table 7, The arrows in the table have no meaning and have not been explained or clarified. An explanation or clarification must be added below the tables to show its meaning.
Authors: Thank you for the comment. It has been corrected (Tables 3, 4 and 5).
Reviewer: 10-The authors used many scientific references in order to increase the modernity of the references. I suggest to add
Authors: Thank you for the suggestion. The references have been added (L728-730, 751-752, 764-765).
Kind regards,
Authors
Reviewer 2 Report
Shaping the physicochemical, functional, microbiological and sensory properties of yoghurts by the plant additives deals with the addition of plant sources to yogurt and their effect on physicochemical, functional, microbiological and sensory properties of yoghurts. The manuscript is well-written. However, suggestions should be consider to improve the quality of the manuscript.
Abstract
Lines 10-14: poorly written
Introduction
Introduction is appropriate. However, authors used more communicable language in many places, such as done (line 83). Done can be replaced with performed
Line 93: including 180 studies. Is it research studies? Please mention it
Table 4 is more general. Since the authors focused on physicochemical, microbiological and sensory characteristics of yoghurts, in-depth details must be provided. Generally, sensory attributes may change extremely and thus authors should present detailed discussion on this. In Table 4. There is not much information on sensory. Authors must illustrate (through Figures) how the yogurt changes by addition of plant additives (before and/or after addition). For example, provide the details of studies, what kind of plant additives used and how it showed effect. Simple decrease and/or increase should not provide a clear message on it.
Sections 4 and 5: Figure containing illustration of work should be provided. For example, Figure X: Microbiological changes in yoghurts; Figure Y: Sensory parameters of yogurt. Show the differences between plant added or normal yogurt.
Table 6
Natural lactose hydrolyzed yoghurt: it Is not from plant origin
Natural yoghurt with the use of bacterial cultures… which bacterial culture?
Natural yoghurt?
3. Physicochemical changes in yoghurts: too many long paragraphs. I suggest authors to divide some reasonable paragraphs that can be easy for a reader to read it
Prophylactic and therapeutic properties of plain yoghurts: Kindly provide Figure summarizing these properties.
Conclusions should be revised to reflect the performed review
Author Response
Dear Reviewer 2,
Thank you for giving us the opportunity to submit a revised version of our manuscript titled “Shaping the physicochemical, functional, microbiological and sensory properties of yoghurts using plant additives” to Foods.
We appreciate the time and effort that you have dedicated to providing your valuable feedback on our manuscript. We are grateful for your comments on our paper. We have made corrections in the text in accordance with the suggestions.
We would also like to emphasize that in the terms of English language, the entire text has been proofread by a sworn translator.
Reviewer: Abstract Lines 10-14: poorly writte.
Authors: Thank you for this suggestion. The lines 10-14 have been changed.
Reviewer: Introduction is appropriate. However, authors used more communicable language in many places, such as done (line 83). Done can be replaced with performed
Authors: Thank you for the suggestion. It has been corrected in the text (L 83).
Reviewer: Line 93: including 180 studies. Is it research studies? Please mention it
Authors: Thank you for the suggestion. It has been corrected in the text (L 93).
Reviewer: Table 4 is more general. Since the authors focused on physicochemical, microbiological and sensory characteristics of yoghurts, in-depth details must be provided. Generally, sensory attributes may change extremely and thus authors should present detailed discussion on this. In Table 4. There is not much information on sensory. Authors must illustrate (through Figures) how the yogurt changes by addition of plant additives (before and/or after addition). For example, provide the details of studies, what kind of plant additives used and how it showed effect. Simple decrease and/or increase should not provide a clear message on it.
Authors: Thank you for the opinion. At such large variety of plant additives used in the research, the results are very divergent. Authors selected exemplary studies and in the form of a timeline presented the maximum addition that was accepted by the testers - the amount that did not affect the deterioration of sensory characteristics. Figure 2 has been created (L 361-364).
Reviewer: Sections 4 and 5: Figure containing illustration of work should be provided. For example, Figure X: Microbiological changes in yoghurts; Figure Y: Sensory parameters of yogurt. Show the differences between plant added or normal yogurt. Thank you for Your opinion, the illustration about microbiological changes has been added - figure 1 - line 281
Authors: Thank you for the comments. Authors believe that the Figures 1 and 2 have clarified and partially summarized this issue (L 268-270 and 361-364).
Reviewer: Table 6 Natural lactose hydrolyzed yoghurt: it Is not from plant origin
Authors: Thank you for the comments. This research was removed from the manuscript.
Reviewer: Natural yoghurt with the use of bacterial cultures… which bacterial culture?
Authors: Thank you for the comment. Information about bacterial cultures has been added – Tables 3 and 5. Tables have been renumbered.
Reviewer: Natural yoghurt?
Authors: Thank you for the question. Information about bacterial cultures has been added – Table 5.
Reviewer: 3. Physicochemical changes in yoghurts: too many long paragraphs. I suggest authors to divide some reasonable paragraphs that can be easy for a reader to read it - line 141
Authors: Thank you for the comment. Part of the text has been moved to Section 7, therefore Section 3 is shorter (L 116-134).
Reviewer: Prophylactic and therapeutic properties of plain yoghurts: Kindly provide Figure summarizing these properties. - Thank you for Your opinion, the illustration about prophylactic and therapeutic properties has beed added - figure 3 line 435
Authors: Thank you for the suggestion. Figure 3 has been added (L 438).
Reviewer: Conclusions should be revised to reflect the performed review
Authors: Thank you for the comment. Conclusion has been changed (L 588-604).
Kind regards,
Authors
Reviewer 3 Report
The presented review aimed to summarize the latest literature on the effect of plant additives in the production of yogurts on their physicochemical, functional, microbiological, and sensory properties. It was found that various additives and their forms are used in the production of yogurts based on the available literature, mainly from the last five years. Nicely presented work, however, the English must be thoroughly revised, and the authors should correct the following:
Page 1. line 35, the word many is unnecessary in this sentence; please remove it.
Page 2, lines 50-51, write the names of lactic bacteria in italic.
Page 2, line 56, check the grammar,
Page 4, lines 72-74, check the grammar,
Page 4, lines 88-92, consider including more of these keywords in keywords of the review below the abstract,
Page 6, considering that carrot and pumpkin are vegetables, correct the sentence to fruit and vegetable additives or plant additives,
Page 7, line 244, specify the types of honey added as a natural sweetener.
Page 9, give an overall conclusion about the differences noticed in sensory features of yogurts, according to Table 3.
Page 10, correct the Table 4 number to Table 3; there are two tables marked as Table 4.
Page 10, at the end of this section, there should be a comparison of the prophylactic and therapeutic properties of plain yogurts and yogurts with plant-based additives.
Author Response
Dear Reviewer 3,
Thank you for giving us the opportunity to submit a revised version of our manuscript titled “Shaping the physicochemical, functional, microbiological and sensory properties of yoghurts using plant additives” to Foods.
We appreciate the time and effort that you have dedicated to providing your valuable feedback on our manuscript. We are grateful for your comments on our paper. We have made corrections in the text in accordance with the suggestions.
We would also like to emphasize that in the terms of English language, the entire text has been proofread by a sworn translator.
Reviewer: Page 1. line 35, the word many is unnecessary in this sentence; please remove it.
Authors: Thank you for the comment. It has been corrected (L 36).
Reviewer: Page 2, lines 50-51, write the names of lactic bacteria in italic.
Authors: Thank you for the comment. It has been corrected (L 51-52).
Reviewer:Page 4, lines 72-74, check the grammar.
Authors: Thank you for the comment. It has been corrected (L 71-76).
Reviewer:Page 4, lines 88-92, consider including more of these keywords in keywords of the review below the abstract
Authors: Thank you for the suggestion. More key words have been added (L 32-33).
Reviewer:Page 6, considering that carrot and pumpkin are vegetables, correct the sentence to fruit and vegetable additives or plant additives
Authors: Thank you for the comment. It has been corrected (L 154).
Reviewer: Page 7, line 244, specify the types of honey added as a natural sweetener.
Authors: Thank you for the comment. The name of honey has been added (L 227-228).
Reviewer:Page 9, give an overall conclusion about the differences noticed in sensory features of yogurts, according to Table 3.
Authors: Thank you for the comment. Figure 1 has been added to present the microbiological changes in the yoghurts with plant additives during the storage, and Figure 2 - to present the sensory acceptability, and Figure 3 for prophylactic and therapeutic properties of yoghurt during regular consuming.
Reviewer:Page 10, correct the Table 4 number to Table 3; there are two tables marked as Table 4.
Authors: Thank you for the comment. It has been corrected.
Reviewer:Page 10, at the end of this section, there should be a comparison of the prophylactic and therapeutic properties of plain yogurts and yogurts with plant-based additives.
Authors: Thank you for the comment. Comparison has been added (L 491-493).
Kind regards,
Authors
Round 2
Reviewer 1 Report
Dear Editor,
The authors made all the required adjustments to the manuscript to make it better, and I now suggest that it be published as is.
Best